# Protocol for the development of a core indicator set for reporting burn wound infection in trials: ICon-B study

Anna Davies,[1] Louise Teare,[2] Sian Falder,[3] Karen Coy,[4] Jo C Dumville,[5] Declan Collins,[6,7] Luke Moore,[8,9] Baljit Dheansa,[10] A Toby A Jenkins,[11] Simon Booth,[12] Riaz Agha,[13] Mamta Shah,[14] Karen Marlow,[3] Amber Young [15,16]

For numbered affiliations see end of article.

**Correspondence to**
Dr Anna Davies;
anna.davies@bristol.ac.uk

## ABSTRACT

**Introduction** Systematic reviews of high-quality randomised controlled trials are necessary to identify effective interventions to impact burn wound infection (BWI) outcomes. Evidence synthesis requires that BWI is reported in a consistent manner. Cochrane reviews investigating interventions for burns report that the indicators used to diagnose BWI are variable or not described, indicating a need to standardise reporting. BWI is complex and diagnosed by clinician judgement, informed by patient-reported symptoms, clinical signs, serum markers of inflammation and bacteria in the wound. Indicators for reporting BWI should be important for diagnosis, frequently observed in patients with BWI and assessed as part of routine healthcare. A minimum (core) set of indicators of BWI, reported consistently, will facilitate evidence synthesis and support clinical decision-making.

**Aims** The Infection Consensus in Burns study aims to identify a core indicator set for reporting the diagnosis of BWI in research studies.

**Methods** (1) Evidence review: a systematic review of indicators used in trials and observational studies reporting BWI outcomes to identify a long list of candidate indicators; (2) refinement of the long list into a smaller set of survey questions with an expert steering group; (3) a two-round Delphi survey with 100 multidisciplinary expert stakeholders, to achieve consensus on a short list of indicators; (4) a consensus meeting with expert stakeholders to agree on the BWI core indicator set.

**Ethics and dissemination** Participants will be recruited through professional bodies, such that ethical approval from the National Health Service (NHS) Health Research Authority (HRA) is not needed. The core indicator set will be disseminated through peer-reviewed publication, co-production with journal editors, research funders and professional bodies, and presentation at national conferences.

**PROSPERO registration number** CRD42018096647.

## Strengths and limitations of this study

► This protocol was co-produced by a multidisciplinary panel with expertise in diagnosing burn wound infection (BWI).
► A systematic review will be undertaken to identify all reported diagnostic indicators of BWI to inform the candidate indicator list.
► We will work closely with multidisciplinary stakeholders who will be identified through professional bodies to participate in the Delphi survey, facilitate dissemination and optimise research impact.
► A UK focus balances limiting the application of the core indicator set with identifying a pragmatic, usable set of indicators for research in the National Health Service (NHS).

## INTRODUCTION

Over 140 000 patients in England and Wales required specialised burn care between 2010 and 2017.[1] Data indicate that the majority of burns treated in the UK are small in size, with a median of between 1.0% and 2.5% of total body surface area affected.[2 3] Patients with burns are susceptible to burn wound infection (BWI) due to the loss of the skin's barrier to microbes and the impact of the burn on the immune system.[4] It is estimated that approximately 10%–20% of patients with burns will develop BWI before healing.[4] These patients may experience delayed wound healing[5] and require regrafting,[6] and can develop sepsis if left untreated. To identify the most effective interventions to prevent, detect and treat BWI, systematic synthesis of evidence from high-quality randomised controlled trials (RCTs) is needed.[7]

Recent systematic reviews have found that while RCTs typically report BWI as a single outcome, there is considerable variation in the indicators used to inform the diagnosis.[8–10] In a Cochrane review of 36 RCTs evaluating the effectiveness of antibiotic prophylaxis in burn patients,[8] 15 studies did not describe the diagnostic indicators used, 14 studies diagnosed burn wound infection using swab culture or biopsy with or without clinical signs, 4 studies used wound signs or clinical observation, 2 studies used systemic signs and one study used chondritis and destruction of cartilige. In another

Cochrane review assessing topical therapy for facial burns,[10] of three studies reporting BWI, three different indicators were used for diagnosis; one study used swab cultures, the second study used qualitative assessment of exudate and cellulitis, and the third study diagnosed BWI if there was chondritis. If the reporting of BWI is based on varying diagnostic indicators, or if the diagnostic indicators used are not reported, the validity of the findings is compromised, limiting the ability of the review to provide good-quality evidence to inform clinical decision-making.

One explanation for this heterogeneity in the reported indicators used to diagnose BWI in research trials is that clinical diagnosis of BWI is challenging; there is no accepted objective method for diagnosing BWI, and diagnosis relies on clinical judgement. Clinical diagnosis is subjective[4] and based on several indicators of infection. These include patient-reported symptoms, clinical wound assessment, systemic signs, the presence of bacteria in a wound, retrospective response to antibiotics and non-specific serum markers of inflammation. Patients typically need to have only some of these symptoms and signs to be diagnosed with, and treated for, BWI. Quantitative bacterial counts from swab cultures are acknowledged as providing only supportive information, as all wounds are colonised with bacteria within 48 hours of injury.[11 12] Diagnosis is further complicated by the fact that many signs used in the diagnosis of BWI, such as fever and tachycardia, are not specific to BWI and are present as part of the normal systemic inflammatory response to burn wound injury.[13]

In the absence of an objective method for diagnosing BWI, consensus statements have been developed with expert stakeholders to support diagnosis of BWI and sepsis. These include the American Burns Association (ABA[14]), the Centers for Disease Control and Prevention (CDC[15]) and the European Wound Management Association (EWMA[16]) consensus statements. However, there are limitations of these statements that preclude their application clinically and in research. The ABA and CDC statements use information gathered from several potential indicators of BWI to result in a composite outcome of BWI. Composite outcomes of BWI have not, and cannot be, tested for diagnostic accuracy due to the absence of an objective reference standard. There are also practical limitations relating to the indicators used to underpin diagnosis. Both the ABA and CDC consensus statements rely on the use of wound biopsy for diagnosis of wound infection. Biopsies are infrequently carried out in the UK because they are costly, invasive, painful and require anaesthesia.[17] The evidence base for the use of quantitative microbiology, such as wound biopsies, for diagnosing clinically relevant BWI is limited.[11 17] Limitations of the EWMA consensus include a lack of reference to systemic signs of infection and inclusion of indicators that, while highly sensitive, are infrequently seen in patients with BWI, for example, ecthyma gangrenosum. For these reasons, these previously developed consensus statements may have limited usefulness for identifying or reporting BWI in UK practice.

Until an evidence-based, objective reference standard for determining the presence of BWI is available, there is a need to develop a consistent approach for reporting the indicators used to inform diagnosis in research. Agreement regarding a minimum (core) set of symptoms, signs and laboratory tests considered to be important in determining the presence of BWI, frequently seen in patients diagnosed with BWI, and assessed as part of normal care will allow standardisation of reporting of BWI in trials.

### Aim

The Infection Consensus in Burns study (ICon-B) aims to achieve the development of a core indicator set to standardise the reporting of symptoms, signs and laboratory tests used for diagnosing BWI in research studies.

## METHODS
### Phase 1: systematic review to identify potential candidate indicators of BWI

A systematic review will be undertaken to identify potential indicators of BWI used in current research to evaluate the effects of interventions on BWI outcomes.

### Identifying relevant studies

Predefined inclusion and exclusion criteria will be used to identify relevant studies. Included studies will be any peer-reviewed, published study or study protocol reporting an evaluation of any intervention, using a randomised or non-randomised trial design or an observational methodology. They will be included if they report the effect of an intervention or treatment for patients with burns, where a BWI outcome is reported in the study methods, results or discussion. Study participants will be any patient with a cutaneous burn injury of any cause and any size. Exclusion criteria are indicated in table 1.

### Search

A search string will be developed based on the above criteria and will comprise synonyms of burns, wound infection, interventions and trials or observational studies (see online supplementary file for example search). It will be applied to five databases: Ovid Medline, Ovid Embase, Cinahl, Cochrane Register of Clinical Trials and Cochrane Register of Protocols. The search will be limited to studies published after 1 January 2010 to ensure that current knowledge and practice regarding BWI are identified. Studies written in languages other than English will be excluded since there is no funding available for translation.

### Selection

The inclusion and exclusion criteria will be applied by one researcher in two stages. These will first be applied to titles and abstracts identified from the search, and then to full texts of studies retained following title and abstract screening. At each stage. a second researcher will review

**Table 1** Inclusion and exclusion criteria for systematic review

| Inclusion criteria | Exclusion criteria |
|---|---|
| ► Trial or observational study assessing the impact of an intervention for patients with burns.<br>► Reports a burn wound infection outcome in the methods, results or discussion. | ► Not written in English.<br>► Not exclusively about burns (eg, trauma population).<br>► Non-human subjects.<br>► Laboratory based, not carried out in clinical setting (eg, exudate or blood samples from humans tested within laboratory).<br>► Does not report a burn wound infection outcome.<br>► Abstract, systematic review or commentary paper.<br>► Not a trial or observational study reporting an intervention to treat burn injuries. |

20% of the screened studies for inclusion, and reasons for exclusion, to ensure reliability of study selection.

### Data extraction

A pro forma will be developed to conduct systematic data extraction with fields relating to study design, intervention/s tested, participant age, gender, burn size and mechanism of injury. Verbatim reports of each indicator for the diagnosis of BWI will be extracted and, where provided, the methods for assessing the indicator (eg, bacterial samples taken using swab) and any standard parameters used to determine the presence of infection (eg, body temperature >38°C, bacterial count of $10^5$ microbes per gram of tissue). One reviewer will conduct all data extraction with a second reviewer extracting data from 20% of studies, to ensure reliability of data extraction. Disagreements will be discussed, and a third reviewer will facilitate decision-making where required. Second-reviewer checking of data extraction will be continued until consistency is achieved.

### Data analysis and reporting

The extracted indicators will be tabulated and examined. Similar indicators will be grouped. A second reviewer will review groupings, and disagreements will be resolved through discussion. Where previously described tools (eg, ABA, CDC and EWMA consensus statements) have been used to diagnose presence of infection, this will be reported. Their incorporated indicators will be deconstructed for inclusion in the long list of candidate indicators.

### Phase 2: reduction of the long list of indicators into a shorter list

The long list of BWI indicators identified in the described systematic review will be reduced by removing duplicates. A steering group will be convened comprising clinicians (doctors, nurses and microbiologists) and researchers with an interest in burns. A face-to-face steering group meeting will be held to review the long list of indicators. Each single indicator or grouping will be assigned a label. The labels will be refined through discussion to achieve a clear definition for each. The labels will be categorised into patient-reported symptoms, clinically observed signs or laboratory tests.

### Phase 3: achieving consensus about BWI indicators using an online Delphi survey

A modified Delphi method will be used, comprising two questionnaire rounds and a final consensus meeting.

An online questionnaire will be developed using REDCap[18] to enable distribution of the questionnaire to a wide stakeholder audience. Before use, the questionnaire will be piloted with a small number of stakeholders from all relevant groups to check understanding and ease of use, as well as data capture and feedback processes.

### Sample

There is no agreed standard sample size for conducting a Delphi survey.[19] Up to 100 stakeholders representing the different staff involved in BWI diagnosis will be recruited, including consultant surgeons working in burn care, junior medical and nursing staff with at least 3 months' experience of burn care, clinical microbiologists working in burn services, and general practitioners and emergency department staff who report that they see at least one patient with suspected BWI per month. Stakeholders will be recruited through mass email to the memberships of appropriate professional groups (eg, the British Burns Association, Care of Burns in Scotland, Royal College of General Practitioners/Emergency Medicine, Association for Clinical Biochemistry and Laboratory Medicine, Healthcare Infection Society and British Society of Antimicrobial Chemotherapy). We will aim to ensure an even distribution of respondents between groups.

### Questionnaire round 1

Participants completing the survey will be asked to rate each of the BWI indicator items on three scales:
1. The importance of the indicator for reporting presence of BWI in research, for example, 'How important is this indicator in diagnosing BWI?'
   Participants will indicate their responses to each indicator on a 9-point Likert-type scale ranging from 1 to 9 with written anchors to indicate 1–3 (not at all important), 4–6 (important but not essential) and 7–9 (very important). Participants may indicate that they have no view about the indicator where appropriate.
2. The frequency with which the indicator is seen in patients with BWI, for example, 'How frequently do

you see this symptom/sign in patients diagnosed with BWI?'

Participants will respond to each indicator on a 9-point Likert-type scale ranging from 1 to 9, with written anchors to indicate 1–3 (never seen), 4–6 (sometimes seen) and 7–9 (seen in most patients diagnosed with BWI). Participants may indicate that they have no view about the indicator where appropriate.

3. Whether the indicator is or can be assessed as part of normal practice for a patient with burns, for example, 'Do you regularly use this indicator to establish the presence or absence of BWI in your day to day practice?'

Participants will respond by indicating whether they would normally assess the indicator as a Yes or No. Participants may indicate that they have no view about the indicator where appropriate.

At the end of the survey, participants will be invited to state any further indicators not included that they believe are important for consideration in the second survey round.

### Calculating consensus about items

The distribution of ratings for each indicator relating to (1) importance and (2) frequency with which the indicator is seen in patients with BWI will be graphed and examined. Percentages will be calculated for the number of participants rating the item 1–3 (not at all important/ never seen), 4–6 (important but not essential/sometimes seen) and 7–9 (very important/seen in most patients diagnosed with a wound infection). Additionally, the percentage will be calculated of participants indicating that they (3) assess the indicator as part of normal practice (% stating yes/no).

### Indicators carried through to the second round of the Delphi

Indicators will only be carried through to the second round of the Delphi where ≥75% of the sample indicate that they assess it as part of normal practice. If this criterion is met, indicators will then be assessed as to whether there is consensus that the item is important and/or frequently seen. Definitions of consensus vary between Delphi surveys; it is accepted that there is a need to balance being so inclusive that a parsimonious core indicator set is not achieved, with using such stringent criteria that important indicators are missed.[20] In several recent core outcome set protocols and studies using Delphi methodology, a threshold of 70% of the sample agreeing that a criterion is important has been used.[21–23] In the current study, a more stringent threshold of ≥75% of the sample falling within the bottom[1–3] or top third of the scale[7–9] of the importance and frequency scales will indicate consensus about the importance/frequency with which an indicator is seen, only where a maximum of 15% of the sample rates it in the opposing third.

Therefore, indicators to be carried forward to the second round of the survey are those where ≥75% of the sample rates it as 7–9 on the Likert-type scale, with

≤15% of the sample rating it as 1–3. Indicators will also be carried through to the second round where there is *no consensus* about their importance or frequency; that is, fewer than ≥75% of the sample rates it in either the top or bottom third of the scale.

### Items will be removed from the indicator list and not carried through to the second round if

1. ≤74% report that they do not assess it as part of their normal practice.
2. There is consensus that the indicator is unimportant OR infrequently seen in patients with BWI, whereby ≥75% of the sample rates it as 1–3, and ≤15% rates it as 7–9 on these scales.

### Round 2

Participants completing the first questionnaire will receive a second survey, comprising the indicators retained from the previous round, using the same item and response format. Additional indicators provided by the respondents in the first round will be incorporated into the second survey round.

In addition, participants will receive tailored feedback showing their own and other stakeholders' responses to each indicator, including the central tendency (eg, mean, median or mode) and distribution of ratings for each item (eg, SD or IQR). Methods for establishing the most effective means for presenting this information will be piloted prior to survey development. It is not anticipated that feedback will be provided indicating item scores according to stakeholder group. However, if there is disparity in responses between groups of stakeholders, this decision will be considered by the study steering group.

Participants will be invited to revisit and, if desired, alter their rating for each item and to restate their ratings for each indicator in relation to (1) importance, (2) frequency and (3) whether it is used for diagnosis.

### Indicators carried through to the stakeholder meeting

Indicators will be taken to the stakeholder meeting if they meet the proceed criteria described above. More stringent criteria may be applied if it is apparent that this process has not reduced the potential indicator list to an acceptable number.

### Phase 4: stakeholder consensus meeting

All survey participants will be invited to a consensus meeting. During this meeting, data will be presented regarding counts/frequency, or central tendency, and the distribution of scores for each indicator relating to (1) whether it is assessed, (2) importance and (3) frequency with which it is observed in patients considered to have BWI. Following review of all indicators, participants will be invited to prioritise indicators for inclusion using anonymous electronic voting software. A final list of indicators will be agreed and a method for reporting the core indicator set in research will be discussed.

## Public and patient involvement

Patients and members of the public were not involved in the development of this protocol, since diagnosis of BWI is made by clinicians.

## Ethics and dissemination

Recruitment of stakeholder participants will take place outside of National Health Service (NHS) organisations and through professional bodies. Therefore, Health Research Authority and research ethics approvals are not required to conduct the survey.

The findings of the systematic review will be presented at relevant burns research-related and wounds research-related conferences and will be written up for publication. The findings of the review will directly inform the items used in the Delphi survey and stakeholder consensus event. Following establishment of a set of core reporting criteria, we will disseminate the findings as a peer-reviewed journal article, and through appropriate professional conferences. We will work with journal editors from relevant journals and will disseminate our findings to relevant funders to increase awareness of the core reporting criteria.

## DISCUSSION

Heterogeneous methods and reporting limits the extent to which evidence synthesis using data from RCTs can provide valid information about the effectiveness of treatments. Systematic literature reviews of treatments for patients with burns demonstrate that there is variation in the indicators used to diagnose BWI in trials, limiting the conclusions that can be drawn about the effectiveness of treatments. In the absence of an objective means of diagnosing BWI, agreement about a set of the most important (core) symptoms, signs and laboratory tests (indicators) for diagnosing BWI will standardise research reporting. This will result in evidence syntheses that provide more valid information about treatment effects. This protocol describes a systematic and pragmatic multiphase approach to developing a core indicator set for BWI, taking account of expert views from the multidisciplinary team involved in BWI diagnosis.

The Delphi methodology is one method to develop consensus among experts on a given topic. It has been used previously to establish consensus between experts about core outcomes to be assessed in various healthcare domains and has been found to improve consistency of outcome reporting between studies.[24] This methodology has also been used to agree key diagnostic criteria for clinical conditions for which there is no objective diagnostic method, including for bone and joint infections in children[25] and hepatic and renal cyst infection.[26] It is important to note that the proposed methodology for developing the core indicator set is not intended to identify those indicators that are most predictive of BWI diagnosis. Rather, its purpose is to identify which indicators multidisciplinary stakeholders agree are important and applicable to routine patient care

in the UK, to standardise the reporting of BWI in research studies. It should also be noted that while the resulting list of indicators is a recommended minimum that should be reported, it will not preclude the reporting of data relating to other indicators relevant to BWI diagnosis, should researchers wish to do so.

The proposed work addresses the practical limitations of previously developed consensus-based criteria for diagnosing BWI (ABA,[27] CDC[15] and EWMA[28]). These consensus statements provide a limited basis for a core indicator set since they include tests that are not currently widely used in NHS practice (eg, wound biopsy, ABA and CDC), and prioritise signs of BWI that are infrequently seen in patients with burns (eg, ecthyma gangrenosum, EWMA[28]). Our proposed study aims not only to take account of whether the indicator is viewed as important in the diagnosis of BWI, but also to ensure that the indicator is likely to be seen and can be assessed in patients with burns. This will ensure that the resulting core indicator set is applicable and relevant to care in the UK, and that data about the indicators can be readily captured for reporting in research studies.

A limitation of the current work is that the literature search to identify current indicators used to diagnose BWI is limited to studies published in English, and that the Delphi survey will only be conducted in the UK. Nonetheless, a global perspective is brought through the identification of international studies published in English. The practical applicability of the core indicator set to NHS practice is balanced against the potential that other indicators than those identified by a UK stakeholder group may be considered of equal or greater importance and usefulness by an international audience.

## Impact and dissemination

To ensure effective uptake of the core indicator set by burn care researchers, early buy-in is essential. By engaging and working closely with the multidisciplinary stakeholder steering group to develop the questions for the Delphi study, and the active participation in the Delphi survey by a wide group of relevant clinical stakeholders, we hope to ensure future uptake of the resulting core indicator set by its intended audience. We aim to publish the core indicator set in a burns-specific journal to reach potential users. Further, we will engage with editors of appropriate journals and the Cochrane Collaboration Wounds Group to facilitate its take-up for trial reporting. Dissemination of the core indicator set and the development work underpinning it described here will be carried out through presentations at appropriate wound-related and burn-related conferences.

## CONCLUSION

The ICon-B study aims to identify a core set of indicators to standardise the reporting of BWI in research studies. The indicators will be those considered to be important for identifying the presence of burn wound infection, frequently observed in patients with BWI and assessed as

part of usual care. A systematic approach to identifying candidate signs, symptoms and tests will be undertaken. The participation of end-user stakeholders to reach consensus about their inclusion will optimise buy-in and impact for the final core indicator set. This will result in consistent recording of data about burn wound infection across studies and will allow the accumulation and comparison of evidence to identify the most effective treatments for patients with burns in relation to BWI.

**Author affiliations**
[1]Centre for Academic Child Health, University of Bristol, Bristol, UK
[2]Department of Microbiology, Chelmsford Hospital, Chelmsford, UK
[3]Burns and Plastic Surgery, Alder Hey Children's NHS Foundation Trust, Liverpool, UK
[4]Centre for Children's Burns Research, University Hospitals Bristol NHS Foundation Trust, Bristol, UK
[5]Division of Nursing, Midwifery and Social work, The University of Manchester, Manchester, UK
[6]Department of Plastic Surgery, Chelsea and Westminster Hospital NHS Foundation Trust, London, UK
[7]Department of Surgery and Cancer, Imperial College London, London, UK
[8]Department of Microbiology, Chelsea and Westminster Hospital NHS Foundation Trust, London, UK
[9]National Institute for Health Research Health Protection Research Unit, Imperial College London, London, UK
[10]Department of Plastic Surgery and Burns, Queen Victoria Hospital NHS Foundation Trust, East Grinstead, UK
[11]Department of Chemistry, University of Bath, Bath, UK
[12]School of Pharmacy and Biomolecular Sciences, University of Brighton, Brighton, UK
[13]Department of Plastic Surgery, Royal Free Hospital NHS Foundation Trust, London, UK
[14]Department of Plastic Surgery, Royal Manchester Children's Hospital, Manchester, UK
[15]Centre for Children's Burns Research, Bristol Royal Hospital for Children, University Hospitals Bristol NHS Foundation Trust, Bristol, UK
[16]Bristol Centre for Surgical Research, University of Bristol, Bristol, UK

**Contributors** AY and AD wrote the paper and conceived the project. LT, ATAJ and LM provided microbiological expertise. SF, KC, DC, BD, SB, RA, MS and KM provided clinical and nursing expertise relating to wound infection diagnosis. JCD provided methodological expertise. All authors critically reviewed, contributed to, and have read and approved the manuscript.

**Funding** This work was supported by an Engineering and Physical Sciences Research Council (EPSRC) Grant, reference: EP/R51164X/1

**Competing interests** None declared.

**Patient consent for publication** Not required.

**Provenance and peer review** Not commissioned; externally peer reviewed.

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
