## [Reviewer comments · BMJ Open]

ARTICLE DETAILS

TITLE (PROVISIONAL)	Protocol for the development of a Core Indicator Set for reporting burn wound infection in trials: The ICon-B study
AUTHORS	Davies, Anna; Teare, Louise; Falder, Sian; Coy, Karen; Dumville, Jo C.; Collins, Declan; Moore, Luke; Dheansa, Baljit; Jenkins, Toby; Booth, Simon; Agha, Riaz; Shah, Mamta; Marlow, Karen; Young, Amber

VERSION 1 - REVIEW

REVIEWER	marc jeschke UofT, Canada
REVIEW RETURNED	29-Aug-2018

GENERAL COMMENTS	While the initiative is excellent i am very skeptical and concerned about this paper. i am not certain why this paper is being written and published. what is the aim and goal of this paper? again, your initiative is exceptional. you need to elaborate why this design of a trial is important for the literature and what does it really add. what is your hypothesis and what is your power analysis? who are your study sites and participants, length, primary and secondary outcomes?
---

REVIEWER	Hajime Matsumura, MD, DMsc, FACS Department of Plastic and Reconstructive Surgery Tokyo Medical University Japan
REVIEW RETURNED	08-Sep-2018

GENERAL COMMENTS	To identify the risk factors of burn wound infection is very important issue. This protocol is well thought out, the feasibility of research is high, and good results are expected if it is implemented.
---

REVIEWER	P JAULT Clinique de La Muette Paris France
REVIEW RETURNED	27-Nov-2018

GENERAL COMMENTS	Thank you for the opportunity of reviewing this paper. The authors propose a method to elaborate a score for a standardization of BWI diagnosis through the analysis of evidence based published articles in burns literature. They submit a proof for a protocol for a development of a Core set for reporting burn wounds infections in Trials: The I Con-B study. Abstract- I think that a clear definition of burn wound infection for the authors is missing. This point is usually unclear in clinical practice, a large definition could be chosen by authors, but some limits should be defined: BWI could be defined by microbiological results, clinical appreciation or by the necessity of a medical intervention. The abstract should be more engaged in this point. In comparison with pulmonary infections there is well defined zone for the colonization, another for infection and a large grey area between by both of them. In BWI, this grey zone is usually large because of subjective assessment of the wound. I'm disagree with the sentence with no objective method to identify BWI...Swabs and biopsies are used worldwide for the diagnosis of BWI. The description of Methodology is clear without any comment. Introduction: The rationale could be improved. One key-point is missing: BWI is more frequently developed in severe burns and is the main cause of death. This point explains why clinicians likely treat with no delay any clinical changes considered as BWI. As swabs are the main bacteriological sample used, a dynamic evolution of the local ecology of the wound is also important. A sterile wound which becomes with positive culture with rare colony of bacteria known 24 hours after the sample was collected, is considered as an infected wound because of the dynamic evolution of the ecology all along the time. This feature is described in the reference n°4, even if the title is confusing with the use of colonization, instead of a dynamic evolution of the ecology of wounds. This point is rarely appreciated in literature, but one of the most clinical important decision key-point. The diagnosis of BWI in 2% burns is never an issue, it's more complex for the third line of infection, or the relapse in a 60% TBSAB. Another point could be highlighted. In BWI the results of bacteriological samples are usually know after 24 hours of culture, there is no direct exam. This is why a wound previously sterile with a positive culture is considered as infected even if there is only few colonies. This complexity of the diagnosis of BWI could be better described in the introduction to be more specific and explain how and why such a complex method is necessary to standardize a complex diagnosis process. Method: In Exclusion (table 1) criteria, can authors develop their choice to exclude studies reporting BWI with other infections. This occurrence is common in severe burns and is always a confusing point in clinical judgement of BWI. The protocol might be more accurate on this specific point.
--

	In phase 1 of the protocol, quality of reviewers should be described. It could be worth to share opinions between biologists, clinicians and plastic surgeon in this first step. Discussion: The relationship between BWI and other infections in burns is in my opinion, a real limitation of the development of an objective selection of criteria in the diagnosis of BWI in severe burns care. The risk is to reduce the selection to local criteria, without any general signs (fever, biological tests evolution).
--	---

VERSION 1 – AUTHOR RESPONSE

1. Reviewers' Comments to Author:

Reviewer 1: Marc Jeschke

While the initiative is excellent i am very skeptical and concerned about this paper. i am not certain why this paper is being written and published. what is the aim and goal of this paper? again, your initiative is exceptional. you need to elaborate why this design of a trial is important for the literature and what does it really add.

Point 1: Aim and goal of the paper

The aim of this paper is to describe a protocol for the systematic, consensus-informed development of a key set of criteria for reporting burn wound infection in studies assessing the effects of any intervention on burn wound infection outcomes. We have identified that to date, systematic reviews of interventions for burns where a burn wound infection outcome is reported have been able to draw only limited conclusions about the effectiveness of these interventions, due to heterogeneity of the criteria applied for diagnosis, and limited reporting of criteria used when reporting data about presence of infection (e.g. Barajas Nava et al., 2013; Avni et al., 2010).

We understand that it is not possible to standardise the complex process of diagnosing burn wound infection, but we believe that standardisation of reporting of the indicators used to diagnose burn wound infection will facilitate evidence synthesis in this area, and advance understanding about the effectiveness of treatments for burns on burn wound infection outcomes. This issue is addressed in paragraphs 2 and 5 of the introduction.

Point 2: Why is this design (of a trial) important for the literature and what does it really add.

There is no agreed gold standard for identifying the presence of clinically-important burn wound infection. Therefore, currently we cannot identify those indicators of burn wound infection that are predictive of diagnosis using the gold standard as the outcome. We therefore need to rely on expert opinion to identify a minimum set of indicators to report burn wound infection, that are agreed by expert professionals, in order to standardise reporting and to allow comparison and collation of study results.

The described method (systematic review and Delphi survey) has been used to develop expert consensus-based agreement about core outcomes in specific healthcare areas (see COMET initiative for references <http://www.comet-initiative.org/>). This methodology has also been previously used to develop diagnostic and reporting criteria for several health conditions, including diagnostic criteria for

bone or joint infections (Mitha et al., 2012). It is therefore an established methodology for facilitating standardisation of reporting in research to support evidence synthesis, including meta-analysis.

The described methodology will capitalise on the expertise of clinicians working in this area to identify those indicators that they believe to be the most important indicators of burn wound infection, and that can be easily recorded and reported in research studies.

Please note this study does not use a trial design.

Point 3: What is your hypothesis and what is your power analysis?

This is not an hypothesis-testing study. The aim of the study is to conduct a systematic review to identify and describe the heterogeneity of burn wound infection indicators used in previous intervention studies for this population, and to conduct a Delphi survey to achieve consensus on which of these are the most important to report in future trials.

In Delphi surveys, and specifically those using the COMET methodology (www.COMET-initiative.org) there are no guidelines for identifying the number of participants required. To ensure that the findings are valid, participants are purposively sampled to ensure that there is adequate representation of important stakeholders in the survey. Typically sample sizes have varied between as few as 40 and into the hundreds, depending on the domain and group of stakeholders with which they have engaged. Therefore, we believe that representation from 100 stakeholders involved in the diagnosis of burn wound infection, and recruitment across all UK burns services, will be adequate to gather sufficient data to give insight into criteria that stakeholders believe are key for reporting in studies assessing burn wound infection outcomes.

Point 4: Who are your study sites and participants, length, primary and secondary outcomes?

Participants in the survey will be professionals, recruited through relevant professional bodies including the British Burns Association, Care of Burns in Scotland, Royal College of General Practitioners/Emergency Medicine, Association for Clinical Biochemistry and Laboratory Medicine, Healthcare Infection Society, British Society of Antimicrobial Chemotherapy. We anticipate referral between practitioners for recruitment into our survey.

Outcomes from the systematic review will be a long-list of infection indicators used to diagnose or report presence of burn wound infection identified from randomised controlled trials where an intervention is tested and its effects on burn wound infection outcomes are reported.

Outcomes from the Delphi survey will be a minimum list of indicators to be reported in future studies, the number of which will be determined by expert stakeholders, and which have been identified through expert consensus as being i) important indicators of burn wound infection, that are ii) frequently seen in patients with burn wound infections, and iii) that can be assessed in routine NHS practice.

Further correspondence with Edward Sucksmith relating to Dr Jeschke's comments 21.01.2019

He did not have any further comments to add beyond stressing that he did not feel that it was important to publish the study protocol of this study. We therefore suggest that you respond by explaining what your study protocol is adding (above and beyond the research paper(s) that will be submitted for publication when the data has been analysed).

We believe that the publication of protocols should not be limited to randomised controlled trials. It is important to write and publish the protocols for both systematic reviews and Delphi-consensus based studies (e.g. Ma, Panaccione, Fedorak et al., 2017; Fish, Sanders, Williamson et al., 2017). Similarly to trial protocols, publication of our proposed systematic review methodology and Delphi survey provides the opportunity to minimise potential bias by stating planned methodology a priori, such that

findings do not influence the eventual methods used or what is reported (Li, Butron, Salman et al., 2016). Furthermore, peer review of the protocol can enable identification of potential flaws that threaten the validity of the research.

Reviewer: 2

Hajime Matsumura, (no comments to address)

Reviewer 3: P JAULT

Abstract-

I think that a clear definition of burn wound infection for the authors is missing. This point is usually unclear in clinical practice, a large definition could be chosen by authors, but some limits should be defined: BWI could be defined by microbiological results, clinical appreciation or by the necessity of a medical intervention. The abstract should be more engaged in this point. In comparison with pulmonary infections there is well defined zone for the colonization, a another for infection and a large grey area between by both of them. In BWI, this grey zone is usually large because of subjective assessment of the wound.

Thank you for your comment. We have altered the abstract text to highlight the complexity of burn wound infection diagnosis and the reliance on microbiology, clinical signs and patient reported symptoms. We would disagree that the necessity of medical intervention, by which we assume the reviewer means commencement of antibiotics, informs diagnosis. We do not consider treatment of suspected burn wound infection an indicator of burn wound infection diagnosis.

The text now reads:

BWI is complex and diagnosed by clinician judgement, informed by patient-reported symptoms, clinical signs, serum markers of inflammation and bacteria in the wound

We believe that the presence of bacteria in any wound does not, in itself, mean the wound is clinically infected and in need of treatment. All wounds will contain bacteria. The microbiological definition of clinical wound infection is the point at which bacterial density increases exponentially and the systemic antibiotics, topical antiseptics and immune clearance have failed to arrest bacterial growth, which was termed by Cutting and Harding in 1994 as critical colonisation (Cutting, K.F.; Harding, K.G. *J. Wound Care* 1994 Jun 2;3(4):198-201). Microbiologically, critical colonisation is the point during bacterial invasion of the wound where genes regulated by Quorum Sensing genes switch on the production of secretory virulence factors (such as delta toxin from *Staphylococcus aureus* and rhamnolipid from *Pseudomonas aeruginosa* [Laabei et al, *PLoS One*, 2014, 9, e87270; Laabei et al *Appl Microbiol Biotechnol* DOI 10.1007/s00253-014-5904-3]. Although the concept of critical colonisation is not universally accepted, we believe that it describes a situation which would correlate with the clinical observation of BWI in that toxin secreting bacteria at high density in the wound matrix cause tissue breakdown (causing pain) and activate immune response, seen clinically as elevated wound temperature and peri-wound erythema.

I'm disagree with the sentence with no objective method to identify BWI...Swabs and biopsies are used worldwide for the diagnosis of BWI.

We agree that it is correct that biopsies or swabs provide evidence of colonisation in wound. However, we disagree that swabs and biopsies are an objective method of determining presence of burn wound infection. Currently there are concerns with cut-offs used to determine a clinically relevant number of bacteria in the wound; these may vary according to service/lab assessing the swab. Furthermore currently stated cut-offs (10⁵ pathogens per gram of tissue) are considered to be arbitrary (Halstead et al., 2018), and there is evidence that these are not well correlated with the presence of clinical signs of infection (see Kallstrom, 2014 for discussion). We believe that swabs provide supporting information to the clinical picture upon which to base diagnosis of burn wound infection.

The description of Methodology is clear without any comment.

Introduction:

The rationale could be improved.

The evidence presented from the identified Cochrane Reviews supports our statement that there is considerable variation in the criteria used to diagnose and report presence of burn wound infections in randomised controlled trials for treatments for this group of patients. Such variation has to date prevented reviewers from drawing conclusions about the effectiveness of reviewed interventions (see paragraph 2 of the introduction). We believe this supports the need to standardise reporting of the indicators of burn wound infection across studies to enable evidence synthesis and meta-analysis.

One key-point is missing: BWI is more frequently developed in severe burns and is the main cause of death. This point explains why clinicians likely treat with no delay any clinical changes considered as BWI. As swabs are the main bacteriological sample used, a dynamic evolution of the local ecology of the wound is also important. A sterile wound which becomes with positive culture with rare colony of bacteria known 24 hours after the sample was collected, is considered as an infected wound because of the dynamic evolution of the ecology all along the time.

This feature is described in the reference n°4, even if the title is confusing with the use of colonization, instead of a dynamic evolution of the ecology of wounds. This point is rarely appreciated in literature, but one of the most clinical important decision key-point.

The diagnosis of BWI in 2% burns is never an issue, it's more complex for the third line of infection, or the relapse in a 60% TBSAB.

As we have discussed above, there is no simple definition of BWI – which is the point of our paper and there is no such thing as a sterile wound, even though bacterial colonization may be at levels which are not easily detectable by classical bacteriology. Wound colonization is of course a dynamic ecological process – indeed we have modelled (in the laboratory) how *Pseudomonas aeruginosa* will out compete *Staphylococcus aureus* in a mixed species wound model.

We are unclear exactly what the reviewer's point is here – I don't think we fundamentally disagree with him.

Another point could be highlighted. In BWI the results of bacteriological samples are usually known after 24 hours of culture, there is no direct exam. This is why a wound previously sterile with a positive culture is considered as infected even if there is only few colonies.

We absolutely agree that patients with larger or deeper burns are more at risk of burn wound infection and we agree that it is usual for clinicians to treat for burn wound infection before the results of bacteriological samples are available, given the risk of death. Since we aim only to identify a minimum set of indicators to be reported in trials for patients with burn injury, we believe that our identified core indicator set will be applicable to all sizes and depths of burn injury.

This complexity of the diagnosis of BWI could be better described in the introduction to be more specific and explain how and why such a complex method is necessary to standardize a complex diagnosis process.

We address diagnostic complexity in paragraph 3 of the introduction, and acknowledge the many indicators that are drawn upon to support diagnosis of infection. However, please note that we are not aiming to report a set of diagnostic indicators, we are interested in identifying those about which data should be reported in future studies to standardise reporting to enable evidence synthesis.

Method:

In Exclusion (table 1) criteria, can authors develop their choice to exclude studies reporting BWI with other infections. This occurrence is common in severe burns and is always a confusing point in clinical judgement of BWI. The protocol might be more accurate on this specific point.

We understand from this comment that Dr Pault is asking whether studies included in the systematic review report infections in addition to burn wound infection and can see the lack of clarity in the criteria we have outlined. I have amended the exclusion criteria table to make this clear. For clarity, we are including studies where burn wound infection is assessed alongside other infections that may be present, such as pneumonia or central line associated infections. We are not including studies where burn wound infection is not assessed.

VERSION 2 – REVIEW

REVIEWER	Jault patrick Clinique La Mulette France
REVIEW RETURNED	17-Mar-2019

GENERAL COMMENTS	Second reading of this manuscript, no comment on this version. This approach is complex, not sure of acceptance by clinicians, should be tried
--